# The Hop2-Mnd1 Complex and Its Regulation of Homologous Recombination

**DOI:** 10.3390/biom13040662

**Published:** 2023-04-10

**Authors:** Hideo Tsubouchi

**Affiliations:** 1Institute of Innovative Research, Tokyo Institute of Technology, Yokohama 226-8503, Kanagawa, Japan; htsubouchi@bio.titech.ac.jp; 2School of Life Science and Technology, Tokyo Institute of Technology, Yokohama 226-8503, Kanagawa, Japan

**Keywords:** Dmc1, homologous recombination, meiosis, Rad51, RecA homologs, synaptonemal complex

## Abstract

Homologous recombination (HR) is essential for meiosis in most sexually reproducing organisms, where it is induced upon entry into meiotic prophase. Meiotic HR is conducted by the collaborative effort of proteins responsible for DNA double-strand break repair and those produced specifically during meiosis. The Hop2-Mnd1 complex was originally identified as a meiosis-specific factor that is indispensable for successful meiosis in budding yeast. Later, it was found that Hop2-Mnd1 is conserved from yeasts to humans, playing essential roles in meiosis. Accumulating evidence suggests that Hop2-Mnd1 promotes RecA-like recombinases towards homology search/strand exchange. This review summarizes studies on the mechanism of the Hop2-Mnd1 complex in promoting HR and beyond.

## 1. Introduction

Homologous recombination (HR) is a universal mechanism found from viruses to humans. HR plays a major role in accurately repairing broken DNA, particularly when both strands are simultaneously broken (i.e., double-stranded breaks or DSBs). Apart from its role in DNA repair in somatic cells, HR is essential for meiosis in most eukaryotic species that undergo sexual reproduction. HR is induced during meiotic prophase, promoting interactions between pairs of homologous chromosomes. This leads to the exchange of genetic material and establishes physical connections between chromosomes, known as chiasmata, which are essential for the accurate segregation of homologous chromosomes at the first division of meiosis.

HR is a highly conserved mechanism, comprised of several common steps (Figure 1) [1,2]. HR is initiated by the presence of a DSB. Upon DSB formation, the ends are resected so that 3′-ended single-strand DNA (ssDNA) is exposed. This ssDNA is used by recombinases to search for intact duplex DNA that is homologous to the ssDNA. The ssDNA invades homologous dsDNA and anneals to the strand complementary to the invading ssDNA while the other strand of the dsDNA is displaced, forming a common early recombination intermediate called the displacement loop (D-loop). The invading 3′ ssDNA serves as a primer for repair synthesis. This strand can be dissociated from the D-loop and then annealed to the other end of DSB, leading to DSB repair without crossing over (i.e., non-crossover). Alternatively, some D-loop intermediates will become a double-Holliday junction (dHJ) if another end of the DSB is annealed to the displaced strand (i.e., second-end capture). dHJ can be dissolved through two pathways. First, a combination of helicase and topoisomerase dissolves dHJ from both ends, thus giving rise to non-crossovers (dissolution). An alternative consequence is that structure-specific nucleases directly resolve HJs, giving rise to crossovers and non-crossovers (resolution). In general, mitotic recombination involves interaction between sister chromatids, which typically leads to non-crossovers. In contrast, interaction and crossing over between homologous chromosomes are promoted during meiosis. The HR pathways described above are somewhat oversimplified, especially when it comes to meiotic recombination. Recent studies using a budding yeast model locus made possible the high-resolution mapping of recombinants arising from a single initiating event [3,4]. The data suggest that both non-crossover and crossover generation involves strand annealing with multiple rounds of strand invasion and that branch migration of Holliday junctions is an integral step in crossover formation.

HR is integrated as an essential part in meiosis [5,6,7,8]. HR progresses simultaneously with dynamic morphological changes of meiotic chromosomes that occur during meiotic prophase, which can be divided into substages based on the morphology of chromosomes (Figure 2). After DNA replication, sister chromatids start condensing along a proteinaceous axis that holds chromatin loops of both sisters (leptotene stage). Simultaneously, induction of DSB formation by meiosis-specific, topoisomerase-like Spo11 protein initiates HR [9]. Then, homologous chromosome axes start aligning with each other and form stretches of synaptonemal complex (SC) (zygotene stage), which eventually encompasses the entire length of the paired homologous chromosomes (pachytene stage) [10]. The SC typically consists of three cytological layers in which a pair of homologous chromosomal axes are closely held by periodically spaced rod-like proteins called transverse filaments. During zygotene, DSBs are processed to form one-ended strand exchange products called single-end invasions, eventually giving rise to dHJs at pachytene. As cells move out of pachytene, the SC starts getting disassembled and, simultaneously, dHJs resolve into crossovers. These crossovers appear as chiasmata at diplotene, linking the homologs.

Not surprisingly, meiotic recombination employs a group of enzymes that play key roles in mitotic recombination [11]. The central player in eukaryotic HR is Rad51, a homolog of RecA, the bacterial homologous recombinase [12]. Rad51 binds nascent 3′ ssDNA overhangs at DSB ends and forms a helical oligomeric complex called the presynaptic complex. Rad51 in the presynaptic complex interrogates intact duplex DNA to search for a similarity between the ssDNA in the presynaptic complex and the target dsDNA. Once homology is identified, Rad51 anneals ssDNA in the presynaptic complex to the complementary strand in the homologous dsDNA, forming a D-loop. In many eukaryotic species, another RecA homolog called Dmc1 is produced specifically during meiotic prophase, playing a central role in meiotic HR. Dmc1 shares similar biochemical properties with Rad51, forming the presynaptic filament, conducting homology searching, and exchanging strands to form D-loops. It remains largely unclear why many eukaryotes require two RecA homologs; indeed, some organisms rely solely on Rad51 for successful meiosis [13].

Both Rad51 and Dmc1 require a host of auxiliary factors involved in different stages of HR [11,14,15]. Resection of DSB ends produces nascent ssDNA, which is immediately occupied by RPA, the eukaryotic ssDNA binding protein. Auxiliary factors known as the mediator facilitate the exchange of ssDNA-bound RPA with RecA homologs, thus promoting the recruitment of RecA homologs to ssDNA [1]. Regulating the oligomeric status of RecA homologs on ssDNA also offers an opportunity to regulate the formation of the presynaptic filament. Other major steps subject to HR regulation are at the formation and stabilization of D-loops. In particular, destabilization or dissociation of D-loops facilitates re-annealing of the extended invading strand to the other end of the DSB, which is not associated with crossing over. Anti-recombinase proteins with helicase activity play this role by dislodging RecA homologs from ssDNA and D-loops [16]. On the other hand, stabilization and extension of a D-loop increases the chance of the displaced strand being captured by the other DSB end, forming a dHJ, thereby increasing the likelihood of crossover formation.

Meiotic recombination is well characterized in yeast models [2,17]. Budding yeast *Saccharomyces cerevisiae* and fission yeast *Schizosaccharomyces pombe* have both Rad51 and Dmc1 [11]. In *S. cerevisiae*, both Rad51 and Dmc1 are essential for successful meiosis. Dmc1 plays a predominant role in meiotic recombination, and its absence almost completely abolishes DSB repair, leaving Rad51 accumulation at the unrepaired DSBs [18,19]. Meiotic progression in the *dmc1* mutant is delayed or arrested at meiotic prophase. In contrast, in *rad51* mutants Dmc1 fails to localize to meiotic chromosomes but *rad51* mutants are capable of low levels of sporulation, although producing inviable spores. In *S. pombe*, the *dmc1* mutant forms viable spores without apparent cell cycle delay, but the frequency of crossing over is substantially reduced [20]. In the absence of Rad51, meiosis is defective and results in dead spores [21]. It has been proposed that in *S. cerevisiae* the primary role of Rad51 in meiosis is not in directly conducting strand exchange, but in recruiting Dmc1 to DSB sites [22]. It is yet to be examined if the same is true in *S. pombe*.

Two major auxiliary factors of Dmc1 have been identified in *S. cerevisiae*: Mei5-Sae3 and Hop2-Mnd1 [23,24,25]. Just like Dmc1, Mei5 and Sae3 are meiosis-specific proteins that form an obligate heterodimer. The absence of these proteins causes practically the same meiotic defects as those caused by the absence of Dmc1: an almost complete defect in meiotic DSB repair and a severe cell cycle delay/arrest at meiotic prophase. Consistent with these phenotypic similarities, the Mei5-Sae3 complex colocalizes with Dmc1 as foci on meiotic chromosomes, and the localization of Mei5-Sae3 and Dmc1 are mutually dependent. These two proteins are conserved in eukaryotic species. Intriguingly, however, their counterpart complex is produced in somatic cells as well as in meiotic cells, serving also as an auxiliary factor of Rad51 [26,27,28,29]. The orthologs of Mei5 and Sae3 are named Sfr1 and Swi5 in fission yeast and mammals, the complex of which is called Swi5-Sfr1. Consistent with its role as a Rad51 auxiliary factor, the *swi5* and *sfr1* mutants show a mild defect in DNA damage repair in both fission yeast and mouse [27,30].

Another meiosis-specific auxiliary factor is the Hop2-Mnd1 complex [25,31]. The absence of Hop2 or Mnd1 leads to a strong cell cycle arrest at meiotic prophase, with DSBs left mostly unrepaired. One of the prominent phenotypes caused by the absence of Hop2 is that non-homologous chromosomes are often found associated during meiotic prophase, suggesting its key role in homolog recognition. Biochemically, the Hop2-Mnd1 complex greatly stimulates the strand-exchange activity of RecA homologs. This review aims to summarize the current understanding of the Hop2-Mnd1 function through its mutant phenotypes, protein behaviors in a cell, biochemical properties, and structure.

## 2. Meiotic Role of the Hop2-Mnd1 Complex

### 2.1. Mutant Phenotypes

*HOP2* was originally identified in *S. cerevisiae* as a meiosis-specific gene whose absence causes a complete cell cycle arrest at meiotic prophase [31]. Notably in *hop2*, many chromosomes form SC between nonhomologous partners (Figure 3A,B). The *hop2* mutant arrests at meiotic prophase with an aberrant accumulation of Dmc1 at unrepaired DSBs, indicating that HR is blocked. Homologous pairing is severely reduced in the *hop2* mutant, which is more severe than that in *dmc1*. Accumulation of nonhomologous synapsis in the *hop2* mutant lead to the proposal that Hop2 prevents synapsis between nonhomologous chromosomes [31]. *MND1* was identified from a whole-genome expression screen for genes required for meiosis and spore formation [32,33]. *MND1* was also identified as a multicopy suppressor of the temperature-sensitive allele of *HOP2* [25]. The *hop2* and *mnd1* mutants essentially share the same phenotypes [23,25,31,33,34,35,36]. Both show a tight cell cycle arrest at meiotic prophase. Meiotic DSBs are poorly repaired, and homolog paring is substantially reduced. Both Rad51 and Dmc1 accumulate aberrantly on meiotic chromosomes in these mutants.

Genetic analysis in *S. cerevisiae* suggests that Hop2-Mnd1 functions in the same pathway as Dmc1. Although meiotic defects of the *dmc1* mutant are very severe, those observed in the *hop2/mnd1* mutants tend to be more severe. In a strain background where the *dmc1* mutant slows down but does not completely arrest, the *hop2* and *mnd1* mutants show an almost complete prophase arrest [38]. Interestingly, a very small fraction of cells form spores in *hop2* and *mnd1*, and their viability is ~10%, which is around the same level as that of the *dmc1* mutant. This is in contrast to the *rad51* mutant where sporulation is much higher (~30%) but spores become mostly inviable. Similarly, DSB repair defect in the *hop2/mnd1* mutants is more severe than that of *dmc1* [38]. By introducing the *dmc1* mutation, however, the meiotic defects observed in the *hop2/mnd1* mutants are suppressed to the level of the *dmc1* single mutant, indicating that the *dmc1* mutant is epistatic to *hop2* and *mnd1*; ~10% of spores become viable in both the *dmc1* and *hop2 dmc1* mutants. Defects caused by the absence of Dmc1, or its meiosis-specific auxiliary factors Mei5 and Sae3, are partially suppressed by the overproduction of Rad51 [23]. The meiotic defects of the *hop2* and *mnd1* mutants are also suppressed by Rad51 overproduction. These observations strongly suggest the specific involvement of Hop2 and Mnd1 in the Dmc1 function. Homologous pairing defects in the double mutants are partially alleviated by introducing *rad51* and/or *dmc1* mutations, leading to the proposal that the meiotic defects in the *hop2/mnd1* mutants are caused by RecA homolog-mediated aberrant activities [38].

In fission yeast *S. pombe*, homologues for Hop2 and Mnd1 were identified: Meu13 and Mcp7, respectively [39,40]. Although, unlike *S. cerevisiae*, *S. pombe* cells do not form the SC, the basic phenotypes of these *S. pombe* mutants are analogous to those found in *S. cerevisiae*. Both mutants show a substantial cell cycle delay during meiosis, which is suppressed by the absence of meiotic DSBs [41]. Homologous pairing and crossing over are heavily compromised in these mutants. The *dmc1* mutant is epistatic to *mcp7* with respect to meiotic cell cycle, spore viability, and crossing over [40].

In mice, *Hop2* knockout mice are infertile in both male and female, consistent with its primary role in meiosis [42]. Spermatocytes from knockout mice arrest prior to the pachytene stage with aberrant accumulation of RAD51 and DMC1, eventually leading to apoptosis. They also accumulate eukaryotic ssDNA binding protein RPA and show persistent γ-H2AX signals. Homologous pairing and formation of the SC are greatly reduced, while formation of chromosome axes is barely affected. Chromosome synapsis is limited, and some advanced cells have SC formed between nonhomologous chromosomes. All these phenotypes are in line with those found in the results obtained from yeast. One striking difference is in the relationship between DMC1 and HOP2. The absence of HOP2 causes a more severe defect in chromosome synapsis than the absence of Dmc1 in mice. The double *Hop2^−/−^ Dmc1^−/−^* mice, however, show a similar synapsis defect to the *Hop2^−/−^* mice. This is in sharp contrast to the observations in budding and fission yeast in which the *dmc1* mutant is epistatic to *hop2* (i.e., the *dmc1 hop2* mutant behaves like *dmc1*). Perhaps *Hop2* has another function in mice, possibly at an earlier stage, before the loading of RecA homologs onto ssDNA.

Hop2-Mnd1 roles have been extensively studied in a plant model *Arabidopsis thaliana*. The *hop2* (the *HOP2* homolog is called *AHP2* in *Arabidopsis*) or *mnd1* mutant plants are sterile [43,44,45,46,47]. They are defective in homologous pairing, and chromosomes look entangled, likely reflecting synapsis between nonhomologous chromosomes. At the stage of prophase corresponding to pachytene and later, chromosome fragmentation becomes evident, indicating their defects in DSB repair. The Hop2-Mnd1 complex likely plays an important role in stabilizing homolog association and preventing connections between nonhomologous chromosomes [48,49]. Unlike the case with *S. cerevisiae* or mice, where *hop2/mnd1* mutants arrest the cell cycle or undergo associated apoptosis, meiosis of *hop2/mnd1* mutants is not arrested in *Arabidopsis*. This is likely because the DNA damage checkpoint seems less stringent in *Arabidopsis* meiosis, which is reminiscent of *S. pombe* meiosis. Meiosis is completed in *dmc1* mutants in both *S. pombe* and *A. thaliana* [20,50]. It is likely that Hop2-Mnd1 functions exclusively with Dmc1 because the *hop2 mnd1 dmc1* triple mutant shows the same phenotype as that of *dmc1* single in which DSBs are repaired through Rad51-mediated intersister HR [51].

In the ciliated protist *Tetrahymena thermophila*, there are two homologs for Hop2 and Mnd1, named Hop2A, Hop2B, Mnd1A, and Mnd1B [52]. Hop2A and Mnd1A are meiosis-specific, while Hop2B and Mnd1B are also expressed in somatic cells. The attempt to knockout *HOP2B* was not successful, suggesting its essential role in this organism. In general, the meiotic defects of the *hop2A* mutant are mild. Meiotic DSBs are likely formed and repaired because γ-H2AX and Rad51 signals appear transiently along the course of meiotic prophase. However, mostly univalents are found at anaphase I, suggesting a defect in crossing over. Homologous pairing is reduced only mildly. The presence of *HOP2B* may account for these relatively mild phenotypes of the *hop2A* mutant. *Tetrahymena* chromosomes at mid meiosis exhibit a unique arrangement where telomeres are clustered at one pole and this bouquet-like configuration is constrained inside the meiotically elongated micronucleus. This unique meiotic configuration may account for the relatively high homologous pairing observed in the absence of Hop2A or Spo11 (and thus no meiotic DSBs).

### 2.2. Localization of Hop2-Mnd1

One unique feature of Hop2 and Mnd1 is their meiotic localization pattern. The meiotic localization of Hop2 and Mnd1, in *S. cerevisiae*, appears to punctate at early prophase, which becomes more intense as chromosomes condense towards the pachytene stage (Figure 3CI–III) [25,31,34]. Hop2/Mnd1 foci do not overlap with Rad51. Importantly, this localization is neither abolished by the absence of meiotic DSBs nor enriched at recombination hot spots (Figure 3CIV,V) [25,31,34]. This is in sharp contrast to the typical localization pattern of RecA homologs and their auxiliary factors. Rad51 and Dmc1 appear as chromosome-associated foci that transiently appear and disappear during the meiotic prophase. This localization pattern coincides with the kinetics of HR maturation. These signals are believed to represent the binding of Rad51 and Dmc1 to the ssDNA produced at DSB ends. This idea is supported by their localization dependency on meiotic DSB formation. Similar localization patterns of Hop2/Mnd1 are observed in *S. pombe* [39,40], *Arabidopsis* [44,45], and the rice *Oryza sativa* [37]. In *Arabidopsis*, although the spatial distribution of Hop2 and Mnd1 shows substantial overlap as foci, the more diffusive Mnd1 signal beyond foci also appears all along the chromosomes, covering a wider area than Hop2. This trend is most prominent at the leptotene and at the nucleolus [44,45]. Super-resolution imaging was used to closely examine Hop2 localization using *O. sativa* (Figure 3D) [37]. OsHop2 is localized to mostly chromatin loops, some of which overlap with chromosome axes and the central region of the SC. Interaction between OsHop2 and Zep1, the transverse filament protein of the SC in rice, is detected on the yeast two-hybrid system, suggesting a possible interplay between Hop2 and the SC [37].

## 3. Structure and Biochemical Properties of the Hop2-Mnd1 Complex

### 3.1. Structure

Hop2 interacts with Mnd1 and forms the Hop2-Mnd1 complex. A crystallographic structure of the *Giardia lamblia* Hop2-Mnd1 complex has been elucidated [53] (Figure 4A,B). Overall, the complex forms a curved, rod-like structure with one end linked to two juxtaposed winged-helix domains (WHD) consisting of the N-terminal residues of each protein, while the other is capped by extra alpha-helices that form a helical bundle structure. Their association is supported by a pair of parallelly arranged alpha-helices. Each helix domain carries three consecutive leucine zippers (LZ1, 2, 3) interrupted by two non-helical regions that form kinked junctions in the complex. The C-terminal leucine zipper of Hop2 and Mnd1 folds back onto LZ3, forming a cap made of alpha-helical bundles, thus designated as LZ3wCH (leucine zipper 3 with capping helices). The WHD domain has a positively charged patch, likely responsible for specific binding to dsDNA binding [54,55,56,57]. On the other hand, the LZ3wCH domain, located on the opposite side, is responsible for binding the Dmc1 nucleoprotein filament.

So far, *G. lamblia* Hop2-Mnd1 is the only complex whose structure has been determined. A combination of cross-linking and mass spectrometry reveals that the *Arabidopsis* Hop2-Mnd1 complex takes a parallel configuration, like that of *G. lamblia* [58]. At the same time, the results suggest the possible coexistence of at least two configurations, open and elongated, and closed and folded. The results also suggest the highly flexible nature of the complex. Interestingly, the mouse HOP2-MND1 complex was proposed to take a V-shape configuration in solution, based on the analysis using a combination of small-angle X-ray scattering and electron microscopy. Two domains for dsDNA-binding (two WHD domains, one for HOP2 and the other for MND1) are separately located on the arms of the V shape. The third DNA binding domain, which preferentially binds ssDNA, is at the C-terminus of HOP2. This is located at the bundled region of the V shape that plays a critical role in interacting with RAD51/DMC1 [56,59]. Variations in the configuration of Hop2-Mnd1 from different species could be due to the rather flexible nature of this complex.

### 3.2. Biochemical Activity

The biochemical properties of mammalian HOP2-MND1 have been extensively studied. Mouse and human HOP2-MND1 preferentially bind dsDNA over ssDNA, interact with both human RAD51 and DMC1, and promote RAD51- and DMC1-driven D-loop formation and strand exchange [54,60,61,62,63,64]. Mouse HOP2-MND1 promotes at least two critical steps of RAD51- and DMC1-mediated reactions: one at the presynaptic filament formation step and the other at the step where the presynaptic filament catches dsDNA to assemble the synaptic complex [62,63]. In the latter step, dsDNA capture was assayed by pulling down presynaptic filaments pre-mixed with dsDNA. Importantly, homology between ssDNA in the presynaptic filament and dsDNA is not a prerequisite for efficient dsDNA capture. Combined with the above-mentioned *G. lamblia* Hop2-Mnd1 structure, LZ3wCH is likely involved in promoting presynaptic filament formation while WHD is responsible for dsDNA binding [53]. HOP2-MND1 stimulates DNA network formation where nucleoprotein filaments containing either Rad51 or Dmc1 associate with multiple dsDNA molecules simultaneously [65]. Single molecular analysis shows that HOP2-MND1 condenses dsDNA [65]. It is interesting to note that *E. coli* RecA can condense DNA into networks while Dmc1 or Rad51, by itself, cannot. Hop2-Mnd1 might provide such an activity to eukaryotic RecA homologs to achieve an efficient homology search. HOP2-MND1 also induces conformational changes of RAD51 that modulate RAD51 binding to nucleotide cofactors and DNA [66]. HOP2-MND1 circumvents the metal ion requirement of RAD51 for ATP binding as well as increases RAD51 binding preference for ssDNA.

Mouse HOP2 promotes D-loop formation by itself, while HOP2-MND1 does not. This raises the possibility that HOP2 can function independently from MND1, and that the intrinsic activity of HOP2 is quenched by MND1 in the complex [60,67]. According to another report, mouse HOP2 showed little D-loop formation activity on its own, but promoted DMC1-driven D-loop formation without MND1 [68]. Indeed, in the *Mnd1* knockout mouse where *Hop2* is intact, spermatocytes show a much higher level of chromosome synapsis, with most meiotically formed DSBs repaired. This raises the possibility that HOP2, either by itself or together with DMC1, supports homologous pairing and synapsis [67]. Further investigation will be necessary to address whether and how HOP2 functions without MND1 in wild type meiosis.

The characterization of the *S. cerevisiae* Hop2-Mnd1 complex initially lagged because the original annotation of the *HOP2* coding sequence (CDS) turned out to be erroneous [36,69]. Nonetheless, the Hop2-Mnd1 complex, using correctly annotated CDS, robustly promotes Dmc1-driven D-loop formation, while it had no effect on Rad51 [69]. Interaction between presynaptic filaments and Hop2-Mnd1 was examined using DNA curtains and single-molecule imaging [70]. Hop2-Mnd1 specifically binds Dmc1-ssDNA filaments, not Rad51 filaments. The association is characterized as quick in binding and slow in dissociation. Such interaction specificity might contribute to the preferential usage of Dmc1 in *S. cerevisiae* meiosis.

*S. pombe* Hop2-Mnd1 promotes Dmc1-driven D-loop formation and strand exchange [64,71]. Interestingly, Hop2-Mnd1 promotes the production of hyper joint molecules, likely a DNA network consisting of associated ss and dsDNA. This is reminiscent of what was observed with mouse Hop2-Mnd1 [65]. *S. pombe* Hop2-Mnd1 interacts with both Dmc1 and Rad51, but it stimulates neither D-loop formation nor strand exchange driven by Rad51. A minor activation of Rad51 was detected when Hop2-Mnd1 was combined with Swi5-Sfr1, a Rad51 auxiliary factor [29,71]. Just like mouse, *S. pombe* Hop2-Mnd1 promotes dsDNA binding by the Dmc1 presynaptic filaments in a homology-independent manner. Hop2-Mnd1 does not stabilize Dmc1 presynaptic filaments in a sense that it does not reduce dissociation of Dmc1 from ssDNA. Similarly, Hop2-Mnd1 is not an efficient mediator of Dmc1 either (i.e., it does not promote the replacement of RPA preassembled on ssDNA with Dmc1). Efficient strand exchange requires Hop2-Mnd1, especially when homology is embedded in the middle of the donor dsDNA, suggesting its critical role in initiating strand exchange [71].

In other organisms, *A. thaliana* Hop2-Mnd1 stimulates Dmc1-mediated D-loop formation [51]. The *Entamoeba histolytica* Hop2-Mnd1 interacts with Dmc1 and also promotes Dmc1-driven D-loop formation and strand exchange [72]. These observations are essentially in line with the reported biochemical properties of Hop2-Mnd1 from other organisms.

## 4. Interplay between the Hop2-Mnd1 Complex and Other Auxiliary Factors

Attempts have been made to reconstitute meiotic recombination using *S. cerevisiae* proteins [73]. Dmc1-driven D-loop formation was reconstituted using Dmc1, Rad51-II3A (a mutant that binds ssDNA but does not execute strand exchange), Mei5-Sae3, Rdh54, and Hop2-Mnd1. When all the proteins are incubated with ssDNA and dsDNA was subsequently added, only a limited amount of D-loop was formed. However, if the initial reaction is supplemented by RPA, D-loop formation was greatly facilitated. A similar effect was seen, even without RPA addition, only if Hop2-Mnd1 was supplemented later, together with dsDNA. RPA likely plays a role in keeping Hop2-Mnd1 away from binding to ssDNA, which is inhibitory to efficient strand exchange.

In *S. pombe*, the relationship between Hop2-Mnd1 and Swi5-Sfr, another major auxiliary factor of Dmc1, has been closely examined [71]. Both auxiliary complexes are highly conserved throughout eukaryotes and share remarkable structural similarity [53]. Despite the similarity, they show substantially different biochemical properties. Both Swi5-Sfr1 and Hop2-Mnd1 strongly promote Dmc1-driven strand exchange. However, Hop2-Mnd1 is more specialized in initiating strand exchange, while Swi5-Sfr1 stabilizes established Dmc1 presynaptic filaments. Together, through their unique contributions, Hop2-Mnd1 and Swi5-Sfr1 synergistically promote Dmc1-driven strand exchange.

## 5. Roles of the Hop2-Mnd1 Complex Outside Meiosis

In *A. thaliana*, Mnd1 is expressed in somatic cells and its transcription is induced upon γ-ray irradiation [74]. Furthermore, the *mnd1* mutant exhibits hypersensitivity to γ-ray, indicating a possible role of Mnd1 in DNA damage repair.

Both human HOP2 and MND1 are expressed in somatic cells [34,75]. Interestingly, HOP2-MND1 is directly involved in a recombination-dependent telomere maintenance pathway called the alternative lengthening of telomeres (ALT) [76]. By artificially introducing DSBs at telomeres in ALT-positive osteosarcoma U2OS cells, two types of telomere movements, random-surveillance and rapid-directional, were induced. These movements likely represent a recombination-dependent telomere maintenance mechanism, which requires RAD51 and the HOP2-MND1 complex. HOP2 is localized as foci near telomeres. Knockdown of *HOP2* or *MND1* strongly reduces telomere clustering and both surveillance and directional movements without affecting RAD51 localization. Mechanisms driven by HOP2-MND1 that promote homologous pairing may play a major role in ALT-based telomere maintenance.

## 6. Hop2-Mnd1 and Human Disease

A mutation in *HOP2* causes a rare, genetically heterogeneous disorder called XX female gonadal dysgenesis (XX-GD), resulting in streak gonads and causing lack of spontaneous pubertal development, causing primary amenorrhea, uterine hypoplasia, and hypergonadotropic hypogonadism [77]. The mutant protein lacks a glutamic acid at 201 (a.k.a., p.Glu201del) and is defective in interaction with RAD51/DMC1 [59]. This correlation suggests a link between the pathogenesis of XX-GD and recombination defects.

Although HOP2 is detected in primary human fibroblasts, it is found highly expressed in various cancer cells, not just ALT but also telomerase-positive types of cancer cell lines [76]. Indeed, there are many cases where genes typically associated with meiosis are found upregulated in cancer cells, suggesting their involvement in carcinogenesis [78]. *HOP2* has been found mutated in familial and early-onset breast and ovarian cancer, as well as fallopian tube cancer patients [79,80]. Some of these mutations lead to alternative splicing, whose products act as dominant negatives, and constitutively expressing the mutant proteins induces tumorigenesis in nude mice [80].

## 7. Perspectives

The combination of genetics, cell biology, and biochemistry using various organisms strongly argues for the pivotal role for the Hop2-Mnd1 complex in activating eukaryotic recombinases, Dmc1 and Rad51. Its primary function is likely executed at the initiation step of strand exchange. Hop2-Mnd1 has a curved rod shape, with one end interacting with the Dmc1/Rad51 presynaptic filaments and the other end binding to dsDNA. It is plausible that Hop2-Mnd1 promotes Dmc1/Rad51-mediated homology search and subsequent strand exchange by mediating the interaction between presynaptic filaments and dsDNA (Figure 3C). The “catching” model assumes that Hop2-Mnd1 associates with presynaptic filaments before homology search. The interaction between the presynaptic filament and dsDNA would be promoted through the dsDNA-catching ability of the N-terminal WHD domain of Hop2-Mnd1. Alternatively, as demonstrated by several cytological studies, Hop2-Mnd1 might primarily function by first binding dsDNA. In this scenario, the presynaptic filament surveying for homology would be “hooked” by the C-terminal LZ domain of Hop2-Mnd1 sitting on dsDNA, facilitating homology search. These two mechanisms are not mutually exclusive.

One unconventional feature of Hop2-Mnd1 as an auxiliary factor of RecA homologs is its localization. RecA homologs and their auxiliary factors are typically recruited to sites of DNA damage. However, Hop2-Mnd1 is localized to chromosomes as many small foci even in the absence of DSB formation and barely overlaps with RecA homolog foci. These observations raise several questions. For example, given that Hop2-Mnd1 is indispensable for meiotic recombination, does HR only occur where Hop2-Mnd1 is localized? Where is Hop2-Mnd1 located, and is there any determinant for its localization? How does Hop2-Mnd1 communicate with RecA homologs during homology search? It is still possible that a relatively small fraction of Hop2-Mnd1, which is below the detection of fluorescence microscopy, works with presynaptic filaments, stimulating homology search/strand exchange, while the foci that appear might represent other functional aspect of Hop2-Mnd1. Interestingly, in an ALT cell line, Hop2 is recruited to a DSB induced near the telomere. It is important to understand where, when, and how Hop2-Mnd1 acts with RecA homologs.

Hop2 and Mnd1 from various organisms form a stable heterodimer whose stoichiometry is 1 to 1. Nevertheless, the phenotypic differences between *Hop2^−/−^* and *Mnd1^−/−^* knockout mice suggest the presence of MND1-independent HOP2 function. Indeed, mouse HOP2 by itself possesses strand-exchange activity. In *Arabidopsis*, Hop2 and Mnd1 localizations do not completely overlap; Mnd1 localization seems to cover a much wider area of meiotic chromosomes than Hop2. This may indicate some degree of functional independence, if any, between Hop2 and Mnd1. Advanced in vivo imaging and high-resolution mapping of the Hop2-Mnd1 localization sites should be able to address some of these questions.

## Figures and Tables

**Figure 1 biomolecules-13-00662-f001:**
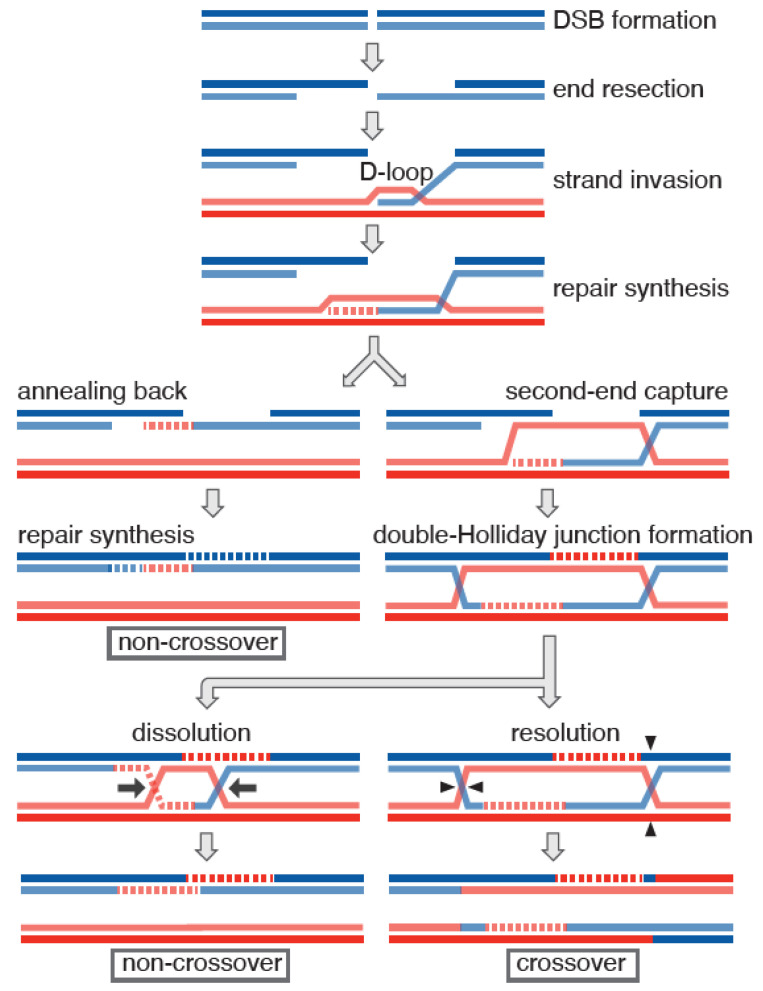
Homologous recombination (HR) steps: The invading 3′-ended ssDNA can anneal to the other end of the DSB, resulting in non-crossovers. Alternatively, annealing of the other DSB end to the displaced strand can lead to the formation of double-Holliday junctions (dHJs). dHJs can be dissolved through either dissolution, forming only non-crossovers, or resolution, forming both crossovers and non-crossovers. These HR pathways could be somewhat oversimplified. Recent studies using budding yeast model locus suggest that both non-crossover and crossover generation during meiosis involves strand annealing with multiple rounds of strand invasion and that branch migration of Holliday junctions is an integral step in crossover formation [3,4].

**Figure 2 biomolecules-13-00662-f002:**
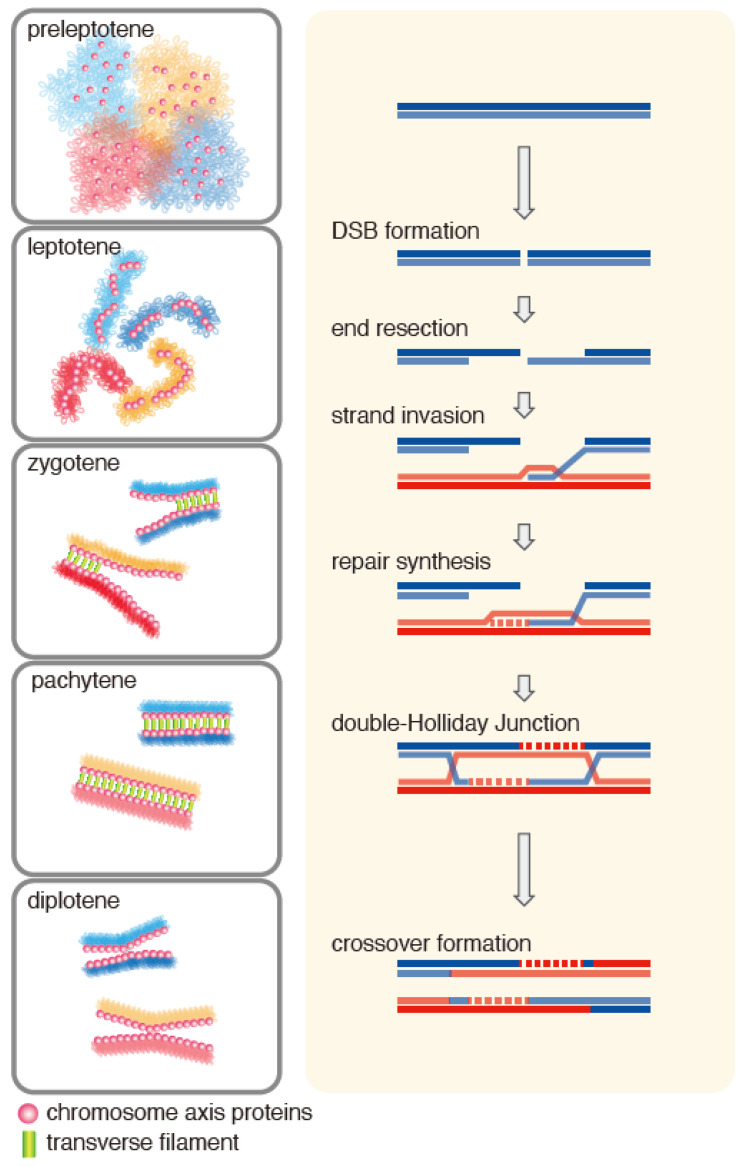
Progression of chromosome morphogenesis relative to the timing of steps involved in homologous recombination during meiotic prophase. For simplicity, the case with only two pairs of homologous chromosomes is shown.

**Figure 3 biomolecules-13-00662-f003:**
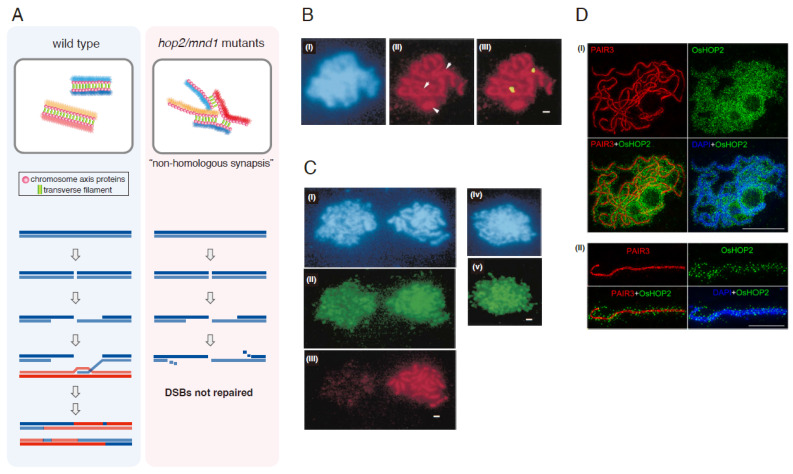
(**A**) Schematic drawing of the typical phenotypes of the *hop2/mnd1* mutants in chromosome morphogenesis and homologous recombination. Synapsis occurs between nonhomologous chromosomes. DSBs successfully initiate and undergo resection, but strand invasion is hindered. (**B**) Meiotic chromosomes of a budding yeast *hop2* strain are stained for (**I**) DNA, (**II**) Zip1, a marker for synapsed chromosomes, and (**III**) both Zip1 and chromosome (**III**). The arrows in (**II**) point to the sites of chromosome (**III**) signals in (**III**). Each unpaired chromosome III is engaged in synapsis. The arrowhead indicates a polycomplex, which is an aggregate of Zip1 often seen in recombination-defective mutants. Scale bar = 1 μm. (From Leu et al. 1998 [31]; reprinted, with permission, from Elsevier). (**C**) Meiotic chromosomes at zygotene (**left**) and pachytene (**right**) are stained for (**I**) DNA, (**II**) Hop2, and (**III**) Zip1. Meiotic chromosomes of the *spo11* strain (thus no meiotic DSBs) are stained for (**IV**) DNA and (**V**) Hop2. Scale bar = 1 μm. (From Leu et al. 1998 [31]; reprinted, with permission, from Elsevier). (**D**) Localization of *Oryza sativa* Hop2 on structured illumination microscopy. (**I**) An *Oryza sativa* meiocyte at pachytene is stained for PAIR3 (meiotic chromosome axis protein), Hop2, and DAPI (DNA). Scale bar = 5 μm. (**II**) A single pachytene chromosome showing most Hop2 foci are localized to the chromatin loops. (From Shi et al. 2018 [37]; reprinted, with permission, from John Wiley and Sons).

**Figure 4 biomolecules-13-00662-f004:**
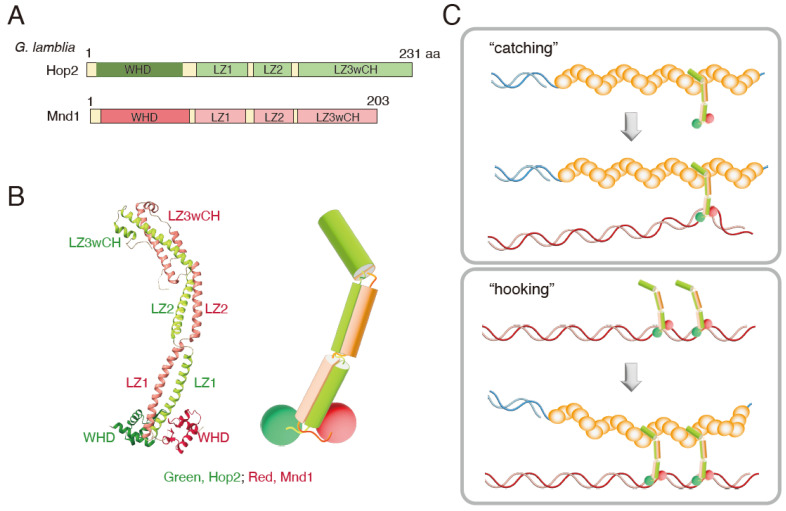
(**A**) Domain organizations of *G. lamblia* Hop2 and Mnd1. WHD, winged-helix domain; LZ, leucine zipper; LZwCH, leucine zipper with capping helices. (**B**) **Left**, crystal structure of the *G. lamblia* Hop2-Mnd1 complex (4Y66, chains D and C). **Right**, schematic drawing of the Hop2-Mnd1 complex. (**C**) Two models of the function of Hop2-Mnd1 in homologous recombination. See the main text for details.

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
