# Peer review of "The Hop2-Mnd1 Complex and Its Regulation of Homologous Recombination"

_biomolecules, 2023, doi:10.3390/biom13040662_

Round 1

Reviewer 2 Report

This review article is well organized and written. It appears up to date and informative.  Other than a small number of grammatical issues I have no suggestions for improvement.

Author Response

This review article is well organized and written. It appears up to date and informative.  Other than a small number of grammatical issues I have no suggestions for improvement.

Thank you so much for reading this article. 

Reviewer 3 Report

The manuscript on DNA repair proteins is well-written, well-structured, and well-illustrated. There are several potential ways to further improve the presentation of the article to readers with a broad background, as specified below.

1) The author's name in the system is spelled as "Hideo Tsubouchi". In the manuscript itself, the author's name is spelled as "Hideo Setouchi". Who is the author of the manuscript? Kindly specify.

2) The paper is dealing with homologous recombination (HR) DNA repair. Several systems are mentioned, including pombi, cerevisiae, Arabidopsis, and mammals (mice). In the introduction, it would be beneficial to mention the key HR proteins, their names in these systems, and their functions. As the Introductory section appears now, it is more of a general description of the HR process, and several proteins are mentioned with a minimal introduction, which makes it hard for non-experts in HR to follow the composition and function of specific proteins during various HR stages.

A table summarizing protein functions and names in various model species could be also beneficial to visualize and conclude this part. 

3) In mice, knockout alleles are usually named in italics with the first letter capitalized, and "-/-" is superscripted. I.e., "dmc1 hop2" double knockouts in mice would be spelled as Dmc1-/-Hop2-/-. Kindly consider using the standard spelling for genes in mice, or justify why the alternative way is preferred. Kindly consult the common ways of gene spelling for various species mentioned in the manuscript.

4) Only 10-12 papers out of 82 references are from the last five years, the majority are older papers. It is beneficial for the literature review to prioritize original works from any period of time, but mainly recent review articles, if review articles are necessary to mention. 

Round 2

Reviewer 3 Report

The author responded to the questions raised during the first round of evaluation and modified the manuscript when suitable.